# Synthesis, Characterization, Cytotoxicity Analysis and Evaluation of Novel Heterocyclic Derivatives of Benzamidine against Periodontal Disease Triggering Bacteria

**DOI:** 10.3390/antibiotics12020306

**Published:** 2023-02-02

**Authors:** Ramasamy Kavitha, Mohammad Auwal Sa’ad, Shivkanya Fuloria, Neeraj Kumar Fuloria, Manickam Ravichandran, Pattabhiraman Lalitha

**Affiliations:** 1Department of Biotechnology, Faculty of Applied Science, AIMST University, Bedong 08100, Kedah, Malaysia; 2Centre of Excellence for Vaccine Development (CoEVD), Faculty of Applied Science, AIMST University, Bedong 08100, Kedah, Malaysia; 3Centre of Excellence for Biomaterials Engineering, Faculty of Pharmacy, AIMST University, Bedong 08100, Kedah, Malaysia; 4Center for Transdisciplinary Research, Department of Pharmacology, Saveetha Institute of Medical and Technical Sciences, Saveetha Dental College and Hospital, Saveetha University, Chennai 600077, Tamil Nadu, India; 5Mygenome, ALPS Global Holding, Kuala Lumpur 50400, Malaysia; 6Department of Biochemistry, Faculty of Medicine, AIMST University, Bedong 08100, Kedah, Malaysia

**Keywords:** periodontal disease, periodontitis, heterocyclics, antibacterial, benzamidine

## Abstract

Periodontal disease (PD) is multifactorial oral disease that damages tooth-supporting tissue. PD treatment includes proper oral hygiene, deep cleaning, antibiotics therapy, and surgery. Despite the availability of basic treatments, some of these are rendered undesirable in PD treatment due to side effects and expense. Therefore, the aim of the present study is to develop novel molecules to combat the PD triggering pathogens. The study involved the synthesis of 4-((5-(substituted-phenyl)-1,3,4-oxadiazol-2-yl)methoxy)benzamidine (**5a-e**), by condensation of 2-(4-carbamimidoylphenoxy)acetohydrazide (**3**) with different aromatic acids; and synthesis of 4-((4-(substituted benzylideneamino)-4H-1,2,4-triazol-3-yl)methoxy)benzamidine *(***6a-b**) by treatment of **compound 3** with CS_2_ followed by hydrazination and a Schiff reaction with different aromatic aldehydes. Synthesized **compounds** were characterized based on the NMR, FTIR, and mass spectrometric data. To assess the effectiveness of the newly synthesized **compound** in PD, new **compounds** were subjected to antimicrobial evaluation against *P. gingivalis* and *E. coli* using the micro-broth dilution method. Synthesized **compounds** were also subjected to cytotoxicity evaluation against HEK-293 cells using an MTT assay. The present study revealed the successful synthesis of heterocyclic derivatives of benzamidine with significant inhibitory potential against *P. gingivalis* and *E. coli*. Synthesized **compounds** exhibited minimal to the absence of cytotoxicity. Significant antimicrobial potential and least/no cytotoxicity of new heterocyclic analogs of benzamidine against PD-triggering bacteria supports their potential application in PD treatment.

## 1. Introduction

Periodontal disease (PD) is a noncommunicable oral inflammatory disease that affects the tissue of the teeth, causing severe damage to the periodontal ligament, leading to tooth loss and reducing the quality of life [1]. PD affects around 11% of the global population [2]. Risk factors such as osteoporosis, metabolic disorder, diabetes, and obesity are strongly linked with chronic PD; however, lifestyle (smoking, alcohol consumption, and poor oral hygiene), poor dietary vitamin D, and calcium intake also play a role [3]. Increasing evidence showed that there is a link between PD and other ailments such as respiratory infections, adverse pregnancy outcomes, cardiovascular diseases, chronic kidney disease, diabetes, cancer, Alzheimer’s, and Parkinson’s disease [4,5,6,7,8]. The onset and progression of PD are associated with the synergy among the organisms found in the microbiota, which then interacts with the host immune defense, leading to severe oral inflammation [9]. The oral microbiota is composed of several organisms that are important in the regulation and protection of the oral cavity against the colonization of non-essential organisms. However, dysregulation of these organisms can cause gingivitis which has the potential to progress to PD [10]. PD is known to be associated with diverse species of bacteria, especially *P. gingivalis* and *E. coli* [11,12]; that are strongly linked to and implicated in the initial onset, progression, and severity of PD and its associated systemic diseases [13,14]. The strategies for prevention and treatment of PD are relatively simple, yet difficult to apply. The reduction of risk factors, the use of probiotic agents, and antioxidants, along with mechanical treatment (scaling) and/or a combination of antibiotics such as metronidazole and amoxicillin or metronidazole and ciprofloxacin can greatly contribute to the reduction or elimination of periodontal associated pathogens [15,16]. Furthermore, they aid the treatment of PD associated systemic diseases [17,18]. Although the systematic adjuvant use of mechanical and combination of antibiotics is the best strategy now, other evidence showed that *P. gingivalis* and its related pathogens are developing resistance to commonly available antibiotics and rendering them less effective by degradation using their virulence factors [19,20,21,22]. Concerning antibiotic resistance, an alternative treatment strategy for periodontal pathogens is the use of synthetic inhibitors. Recent evidence showed that synthetic molecules have the potential to ease the burden of oral infections caused by *P. gingivalis* with no significant cytotoxicity observed [23,24]. Facts suggest that incorporation of heterocyclic groups into the organic moieties enhances their biological potential [25,26]. A study reported that benzamidine and its derivatives displayed inhibition against gingipains, a major virulence factor produced by *P. gingivalis* [27]. Our previous study showed that benzamidine and its derivatives (ester, hydrazides, and Schiff bases) inhibit *P. gingivalis* and its associated pathogens [12]. Recent studies showed effectiveness of oxadiazoles and triazoles against periodontitis triggering pathogens [28,29,30]. To combat antibiotic resistance, oxadiazoles have been used due to their sensitive antimicrobial activity [31]. Hence, based on the severity of PD, associated pathogens and their resistance, potential of benzamidine analogs against PD, and the enhancement of inhibitory potential by incorporation of oxadiazoles and triazoles groups in different chemical moieties, researchers are motivated to perform the synthesis, characterization, cytotoxicity analysis, and evaluation of novel heterocyclic derivatives (oxadiazoles and triazoles) of benzamidine against periodontal-disease-triggering bacteria. In the continuation of a previous study, our present study highlights that the synthesis of new oxadiazole and triazole derivatives of benzamidine analogs possess high inhibition potential against triggering bacteria, which makes them the forefront of potential PD treatment.

## 2. Materials and Methods

### 2.1. General Information

The reagents, solvents, and chemicals used for the synthesis of compounds in the present study were acquired from Sigma-Aldrich Co. (St. Louis, MO, USA), HmbG^®^ Chemicals, Hamburg, Germany, Friendemann Schmidt Chemical, Washington, DC, USA, Merck KGaA (Darmstadt, Germany), and Qrec Chemicals, Rawang, Malaysia. The Ashless Whattman No. 1 filter paper was used for filtration. To verify compounds’ purity, the open capillary tube method was used. The melting points of all synthesized compounds were determined using SMP11 Analogue apparatus. The compounds’ characterization was recorded by ^1^H-NMR and ^13^C-NMR (NMR 700 MHz ASCEND™ spectrometer) using deuterated DMSO solvent, on a δ value scale as the downfield chemical shift in ppm against tetramethylsilane (TMS). The NMR signals are stated as s, single; d, doublet; t, triplet; m, multiplet. The IR of synthesized compounds was recorded using a Jasco ft/ir-6700 instrument in a wavelength range of 400–4000 cm^−1^. The analysis of mass spectra was recorded from a Direct Infusion IonTrap MS Full Scan (Thermo Scientific Q Exactive HF-X hybrid quadrupole-Orbitrap mass spectrometer, Waltham, MA, USA). Elemental analysis was performed on a Perkin Elmer 240 B and 240 C. Elemental analysis (C, H, N), indicated by employing element symbols, was within ±0.4% of theoretical values. The purity of compounds and monitoring of reactions were assessed by TLC on aluminum sheets with silica gel 60 F254 (0.2 mm) (Merck Millipore, Darmstadt, Germany) using methanol: chloroform (0.3: 1.7) as a solvent system in a UV chamber using a SPRECTROLINE^®^ CM-26 UV viewing chamber. The 4-hydroxybenzenecarboximidamide analogs were synthesized as per the protocol given by previous authors with slight modifications [32,33,34,35,36].

### 2.2. Synthesis

#### 2.2.1. General Procedure for the Synthesis of 4-((5-(Substituted-phenyl)-1,3,4-oxadiazol-2-yl)methoxy)benzamidine (**5a-e**)

To synthesize the oxadiazole derivatives of benzamidine (**5a-e**), an equal molar concentration of **compound** (**3**) (0.02 M) and 3-phenoxy benzoic acid was dissolved in 10 mL of phosphoryl chloride and refluxed for 8 h. At the end, the mixture was cooled, washed with ice, filtered, and recrystallized to obtain the pure **compound 5a**. The synthetic scheme for synthesis of **compound 5a-e** is given in Figure 1. During the experiment, anhydrous reaction conditions were maintained, and the recrystallization was done using methanol and activated charcoal. The synthesized **compound 5a** was further characterized based on the spectrometric data (Appendix A). Similarly, other **compounds 5b-e** were synthesized, purified, and characterized.

##### 4-((5-(4-Chlorophenyl)-1,3,4-oxadiazol-2-yl)methoxy)benzamidine (**5a**)

White crystalline (Yield 73%, m.p. 184 °C); IR (KBr, cm^−1^): 3051 (Aromatic C–H), 2978 (Aliphatic H–C), 1681 (C=N), 1591, 1425 (Aromatic C=C), 1238, 1066 (C–O–C of oxadiazole ring); ^1^H-NMR (DMSO-*d*_6_, ppm) δ: 3.21 (s, 2H, O–CH_2_), 3.71 (brs, 2H, NH_2_), 7.25–7.79 (m, 8H, Ar–H), 8.41 (s, 1H, C=NH); and ^13^C–NMR (DMSO, ppm) δ: 68.17 (CH_2_), 122.93, 128.35, 128.88, 129.29, 130.79, 130.95, 131.432, 133.62, 138.86 (Ar-C), 167.47 (C=N); Mass (m/z): Calcd. 328.75, found 328.4; Anal. Calcd. for C_16_H_13_ClN_4_O_2_: C, 58.45; H, 3.99; N, 17.04%, Found: C, 58.53; H, 3.91; N, 17.12%.

##### 4-((5-(2-Chlorophenyl)-1,3,4-oxadiazol-2-yl)methoxy)benzamidine (**5b**)

Beige crystalline (Yield 70%, m.p. 124 °C); IR (KBr, cm^−1^): 3065 (Aromatic C–H), 2998, 2888 (Aliphatic C–H), 1682 (C=N), 1569, 1474 (Aromatic C=C), 1265, 1043 (C–O–C of oxadiazole ring); ^1^H-NMR (DMSO-*d*_6_, ppm) δ: 3.21 (s, 2H, O–CH_2_), 3.72 (brs, 2H, NH_2_), 7.46–7.83 (m, 8H, Ar-H), 8.12 (s, 1H, C=NH); ^13^C-NMR (DMSO, ppm) δ: δ 68.18 (O–CH_2_), 122.51, 127.64, 131.01, 131.17, 131.50, 131.97, 132.07, 132.93 (Ar-C), 167.24 (C=N); Mass (m/z): Calcd. 328.75, found 328.5; Anal. Calcd. for C_16_H_13_ClN_4_O_2_: C, 58.45; H, 3.99; N, 17.04%, Found: C, 58.58; H, 3.93; N, 17.16%.

##### 4-((5-(2-Fluorophenyl)-1,3,4-oxadiazol-2-yl)methoxy)benzamidine (**5c**)

Light brown crystalline (Yield 75%, m.p. 138 °C); IR (KBr, cm^−1^): 3051 (Aromatic C–H), 2917 (Aliphatic C–H), 1675 (C=N), 1508, 1426 (Aromatic C=C), 1292, 1067 (C–O–C of oxadiazole ring); ^1^H-NMR (DMSO-*d*_6_, ppm) δ: 3.20 (s, 2H, CH_2_), 3.49 (brs, 2H, NH_2_), 7.30–8.01 (m, 8H, Ar–H), 8.01 (s, 1H, C=NH); ^13^C-NMR (DMSO, ppm) δ: 68.44 (O-CH_2_), 115.99, 117.20, 122.63, 123.90, 127.79, 128.77, 129.87, 131.40, 132.35, 134.56 (Ar–C), 164.67 (C=N); Mass (m/z): Calcd. 312.3, found 312.1; Anal. Calcd. for C_16_H_13_FN_4_O_2_: C, 61.53; H, 4.20; N, 17.94%, Found: C, 61.61; H, 4.25; N, 17.88%.

##### 4-((5-(4-Aminophenyl)-1,3,4-oxadiazol-2-yl)methoxy)benzamidine (**5d**)

Brown crystalline (Yield 70%, m.p. 134 °C); IR (KBr, cm^−1^): 3229 (N–H), 3065 (Aromatic C–H), 2918 (Aliphatic C–H), 1653 (C=N), 1593, 1407 (Aromatic C=C), 1243, 1062 (C–O–C of oxadiazole ring); ^1^H-NMR (DMSO-*d*_6_, ppm) δ: 3.22 (s, 2H, O-CH_2_), 3.70–4.20 (brs, 4H, NH_2_ and Ar–NH_2_), 7.03–8.11 (m, 8H, Ar-H), 8.12 (s, 1H, C=NH); ^13^C-NMR (DMSO, ppm) δ: 68.11 (O–CH2), 113.06, 116.17, 119.70, 120.03, 123.24, 130.09, 130.62, 131.82 (Ar–C), 167.19 (C=N); Mass (m/z): Calcd. 384.3, found 384.10; Anal. Calcd. for C_16_H_15_N_5_O_2_: C, 62.13; H, 4.89; N, 22.64%, Found: C, 62.22; H, 4.93; N, 22.71%. 

##### 4-((5-(3,5-Dinitrophenyl)-1,3,4-oxadiazol-2-yl)methoxy)benzamidine (**5e**)

Yellowish brown crystalline (Yield 75%, m.p. 130 °C); IR (KBr, cm^−1^): 3051 (Aromatic C–H), 2919 (Aliphatic C–H), 1680 (C=N), 1589, 1474 (Aromatic C=C), 1310 (N-O), 1242, 1042 (C–O–C of oxadiazole ring); ^1^H-NMR (DMSO-*d*_6_, ppm) δ: 3.16 (s, 2H, O–CH_2_), 3.71 (brs, 2H, NH_2_), 7.41–7.78 (m, 7H, Ar–H), 8.02 (s, 1H, C=NH); ^13^C-NMR (DMSO, ppm) δ: 67.98 (O–CH_2_), 122.12, 123.26, 127.7, 128.07, 129.21, 130.56, 131.06, 132.01, 133.28, 134.55 (Ar–C), 167.23 (C=N); Mass (m/z): Calcd. 309.32, Found 309.20; Anal. Calcd. for C_16_H_12_N_6_O_6_: C, 50.01; H, 3.15; N, 21.87%, Found: C, 50.12; H, 3.11; N, 21.79%.

#### 2.2.2. General Procedure for the Synthesis of 4-((4-3-Phenoxybenzylideneamino)4-4-nitrobenzylideneamino)-4H-1,2,4-triazole-3-yl methoxy)benzamidine (**6a-b**)

To synthesize the **compound 6a,b**, a mixture of **compound 3** (0.1 M), potassium hydroxide (0.15 M), and CS_2_ (0.15 M) in absolute ethanol was stirred for 18 h. To the resulting solution, 250 mL of anhydrous ether was added to precipitate potassium dithiocarbazinate. The 0.02 M of dithiocarbazinate was hydrazinated with 0.04 M of hydrazine hydrate. The hydrazinated product was treated with different aromatic aldehydes separately in equimolar concentration. The synthesized **compounds** were recrystallized using absolute ethanol to offer pure **compounds 6a,b**. The synthetic scheme for synthesis of **compound 6a-b** is given in Figure 1. During the experiment, anhydrous reaction conditions were maintained, and the recrystallization was done using methanol and activated charcoal. The synthesized **compound 6a** was further characterized based on spectrometric data (Appendix A). Similarly, **compound 6b** was also synthesized, purified, and characterized.

##### 4-((4-(3-Phenoxybenzylideneamino)-4H-1,2,4-triazole-3-yl)methoxy)benzamidine (**6a**)

Yellow crystalline (Yield 85%, m.p 193 °C); IR (KBr, cm^−1^): 3034 (Aromatic C–H), 2918 (Aliphatic C–H), 1687 (C=N), 1481, 1447 (Aromatic C=C); ^1^H-NMR (DMSO-*d*_6_, ppm) δ: 3.35 (s, 2H, O–CH2), 7.06 (s, 1H, S–H), 7.17–7.71 (m, 22H, Ar–H), 9.29 (s, 1H, N=CH), 9.99 (s, 1H, C=NH); ^13^C-NMR (DMSO, ppm) δ: 67.89 (O–CH2), 117.82, 118.59, 119.75, 119.81, 123.23, 124.52, 124.58, 124.75, 124.89, 125.22, 130.71, 130.77, 130.83, 131.51, 133.30, 138.42, 163.33 (Ar–C), 157.75 (C=N), 167.38 (1H, C=NH), 193.01 (N=C-S); Mass (m/z): Calcd. 624.71, Found 624.20; Anal. Calcd. for C_36_H_28_N_6_O_3_S: C, 69.21; H, 4.52; N, 13.45%, Found: C, 69.18; H, 4.59; N, 13.51%.

##### 4-((4-(4-Nitrobenzylideneamino)-4H-1,2,4-triazole-3-yl)methoxy)benzamidine (**6b**)

Yellow crystalline (Yield 88%, m.p 180 °C); IR (KBr, cm^−1^): 3052 (Aromatic C–H), 2919 (Aliphatic C–H), 1703 (C=N), 1537, 1444 (Aromatic C=C); ^1^H-NMR (DMSO-*d*_6_, ppm) δ: 3.36 (s, 2H, O–CH2), 7.06 (s, 1H, S–H), 8.16–8.42 (m, 12H, Ar–H), 9.29 (s, 1H, N=CH), 9.98 (s, 1H, C=NH); ^13^C-NMR (DMSO, ppm) δ: 68.01 (O–CH2), 124.73, 125.32, 127.12, 131.11, 131.57, 135.28, 133.67, 136.59, 138.45, 139.17, 139.82, 140.54 (Ar–C), 151.09 (C=N), 167.82 (C=NH), 192.79 (N=C-S); Mass (m/z): Calcd. 530.52, Found 530.20. Anal. Calcd. for C_24_H_18_N_8_O_5_S: C, 54.34; H, 3.42; N, 21.12%, Found: C, 54.29; H, 3.39; N, 21.08%. 

### 2.3. Determination of Antimicrobial Activity

In the present study, the micro-broth dilution method was used to determine the inhibition susceptibility of synthesized **compounds** against *P. gingivalis* (ATCC 33277) and *E. coli* (ATCC 25922). The strain of *P. gingivalis* and *E. coli* were obtained from ATCC. The bacteria were cultured in blood-enriched tryptic soy agar (eTSA) (Merck KGaA, Darmstadt, Germany), supplemented with sterile filtered 5% L-cysteine (Bio-Basic, Markham, ON, Canada), 1% dithiothreitol (Sigma Life Sciences, Burlington, MA, USA), and 0.5 mg/mL vitamin K (Sigma Life Sciences, Burlington, MA, USA) with an adjusted pH of 7.4 [37]. As per CLSI guidelines, the micro broth dilution method was used to determine the minimum inhibition concentration (MIC) of *P. gingivalis*. The synthesized **compounds** were diluted in two-fold serial dilution, starting with the highest concentration at 500 µg/mL, and the lowest concentration at 7.8125 µg/mL. Ampicillin was used as a control (Akum Drugs and Pharmaceuticals, New Delhi, India), with final concentrations of 250 µg/mL to 1.6 µg/mL. All of these dilutions were carried out aseptically. To inoculate bacterial culture for MIC, the 0.5 McFarland standard was used (1.5 × 10^8^ CFU/mL) [38,39]. To the microtiter plate, an equal volume of 1.5 × 10^8^ CFU/mL of *P. gingivalis* was added, excluding only the negative control. The microtiter plates were incubated in an anaerobic jar (Oxoid, Winchester, UK) supplemented with a gas pack (Merck KGaA, Darmstadt, Germany) that generate 90% N_2_, 5% CO_2_, and H_2_ and a gas indicator (Thermo Fisher Scientific, Waltham, MA, USA) for 46 h at 37 °C.

Cation-adjusted Mueller–Hinton broth (CAMHB) and agar (CAMHA) (HiMedia, Mumbai, India) were used for *E. coli* MIC evaluation. The MIC of *E. coli* was determined using the same method as that of *P. gingivalis*. The final concentration of the ampicillin was 250 µg/mL to 1.6 µg/mL (CSC Pharmaceuticals, Mumbai, India). The microtiter plates were incubated at 37 °C in aerobic conditions for 18 h.

To determine *P. gingivalis* minimum bactericidal concentration, MIC results of the clear wells of samples where there was no visible bacterial growth were aseptically plated on eTSB agar and incubated in an anaerobic jar at 37 °C with a gas indicator and gas pack for 46 h. The MBC of *E. coli* was determined by plating the MIC results of the clear wells on CAMHA and incubating for 18 h at 37 °C according to the guidelines given by CLSI. After incubation, MBC was recorded as the lowest concentration of a **compound** with no visible growth of bacteria with agar clarity, the same as that of the negative control. All experiments were performed in triplicate.

#### 2.3.1. Cell Viability Assay

MTT is the most common cell viability assay used and it depends on the conversion of substrate to a chromogenic product by live cells. This assay involves the conversion of the water-soluble MTT [3-(4,5-dimethylthiazol-2-yl)-2,5-diphenyltetrazolium bromide] to an insoluble formazan by the action of mitochondrial reductase. The solubilized formazan concentration is then determined by an optical density at 570 nm [40]. To determine cell viability, HEK 293 cells obtained from ATCC were revived and cultured using DMEM (Dulbecco’s modified Eagle medium) supplemented with 5% fetal bovine serum (FBS) (Sigma life science, Burlington, MA, USA) and 1% antibiotic (GIBCO, Waltham, MA, USA) and incubated at 37 °C, with 5% CO_2_, and relative humidity of about 95% (Heal Force/HF90, Hong Kong, China). 

#### 2.3.2. Cell Counting

Cells were counted using the hemocytometer (Hirschmann Laborgerate, Darmstadt, Germany) counting technique. Here, cells were washed with PBS (First base, Axil Scientific, Singapore), treated with trypsin (Sigma life science, Burlington, MA, USA), and incubated at 37 °C to detach them from the flask surface. After trypsinization, 0.1 mL of cells were added to 0.9 mL of 0.2% trypan blue (Sigma life science, Burlington, MA, USA) in a sterile microcentrifuge tube. A 10 μL sample of stained cells were loaded into both sides of the chamber of the hemocytometer, covered with microscopic cover glass, and cells were viewed under the inverted microscope (Olympus/CK40-F200, Shinjuku, Japan). The viable cells were not stained with trypan blue, whereas the dead cells were stained [41]. Using Equations (1) and (2), the total viable cells were calculated:(1)$$\mathrm{Total}\;\mathrm{number}\;\mathrm{of}\;\mathrm{viable}\;\mathrm{cells}=\frac{\mathrm{Total}\;\mathrm{number}\;\mathrm{of}\;\mathrm{cells}\;\mathrm{from}\;4\;\mathrm{grids}}{4}$$
(2)$$\mathrm{Number}\;\mathrm{of}\;\mathrm{live}\;\mathrm{cells}\;=\mathrm{Total}\;\mathrm{number}\;\mathrm{of}\;\mathrm{viable}\;\mathrm{cells}\;\times \mathrm{Dilution}\;\mathrm{factor}\;\times \;10^{4}/\mathrm{mL}$$

#### 2.3.3. Cell Treatment

After 24 h of incubation, the counted cells were treated with different concentrations (50–7.8125 μg/mL) of the synthesized **compounds**. All seeded cells were treated with synthesized **compounds** except the controls, and cells were further incubated until MTT analysis [42].

#### 2.3.4. 3-[4,5-Dimethylthiazol-2-yl]2,5-diphenyl Tetrazolium Bromide (MTT) Assay

A 20 µL sample of MTT reagent (0.5 mg/mL) (Sigma life science, Burlington, MA, USA) in PBS was added to each well, including the controls, and the plates were covered with aluminum foil paper and incubated at 37 °C for 4 h. After incubation, the cells were treated with MTT detergent (DMSO) (Sigma life science, Burlington, MA, USA) and further incubated for 1 h. After 1 h of incubation, the sample OD was measured at 570 nm with the reference of 630 nm using Infinite 200 PRO (Tecan Microplate Reader, Mannedorf, Switzerland). A triplicate experiment was carried out for all synthesized **compounds**. Below is the formula used for calculating the percentage (%) of cell viability:(3)$$\mathrm{Cytotoxicity}\;\left(\%\right)=\frac{\mathrm{Sample}\;\mathrm{Absorbance}\;\left(\mathrm{mean}\right)}{\mathrm{Control}\;\mathrm{Absorbance}\;\left(\mathrm{mean}\right)}$$

#### 2.3.5. Statistical Analysis

GraphPad Prism software version 5 (GraphPad Software, Inc., San Diego, CA, USA) was used to analyze cytotoxicity statistical data. One-way analysis of variance (ANOVA) followed by Dunnett’s post-hoc test to determine the source of significant difference between the groups using SPSS software (IBM SPSS Statistics, Version 25). Results are presented as mean ± standard error of experiment performed in triplicate.

## 3. Results and Discussion

### 3.1. Chemistry

In a previous study, the authors of the present study, described the synthesis of **compounds 2**, **3**, and **4a-c**, that involved preparation of ethyl-2-(4-carbamimidoylphenoxy)acetate 2, by esterification of 4-hydroxybenzenecarboximidamide (**1**), followed by hydrazination to form 2-(4-carbamimidoylphenoxy)acetohydrazide (**3**), which was further treated with different aromatic aldehydes to offer N-(substituted benzylidene)-2-(4-(N-(4-ydroxybenzylidene)carbamimidoyl)phenoxy)acetohydrazide (**4a-c**) [12]. 

In the current study, **compound 3** was subjected to different types of reactions. In one part of the experiment, **compound** (**3**) was treated with different aromatic acids (4-chlorobenzoic acid, 2-chlorobenzoic acid, 4-flurobenzoic acid, 4-aminobenzoic acid, and 3,5-dinitrobenzoic acid) in the presence of POCl_3_ to offer new oxadiazoles derivatives of benzamidine (**5a-e**). The stated experiment involved cyclo-condensation reaction of aromatic acids with hydrazide (**3**) in the presence of POCl_3_ to form **compound 5a-e**. The physical and chemical properties of newly synthesized **compounds** in the present study are also supported by other investigations [43,44]. Whereas, in another part of the experiment, **compound** (**3**) was treated with carbon disulfide in the presence of potassium hydroxide to offer potassium dithiocarbazinate, which was further subjected to hydrazination followed by treatment with different aromatic aldehydes to offer **compounds** (**6a**,**b**) [45,46,47]. The synthetic scheme for all new **compounds 5a-e** and **6a-b** is given in Figure 1. 

Figure 1 shows synthetic route of novel analogs obtained in this study. The purity of synthesized **compounds** was determined based on melting point, single spot TLC (thin-layer chromatography) pattern, and CHN analysis. In the present study, the spectrometric analysis of synthesized **compounds** using mass spectrometry, FTIR, ^1^H, and ^13^C-NMR confirmed the structure of **compounds 5a-e** and **6a-b**. The successful synthesis of **compound 5a-e** was confirmed based on the presence of the characteristic IR bands at 1042–1295 (C–O–C of oxadiazole ring), disappearance of ^1^H-NMR signals at 8.27, appearance of extra 13C-NMR signals for aromatic carbons raging between 113 and 166, and the appearance of a mass spectrum ion peak ranging between 309 and 384 confirmed the structure of the synthesized **compounds 5a-e**. The successful synthesis of the **compound** (**6a-b**) was confirmed based on the appearance of the characteristic IR bands at 1687 and 1703 (C=N), ^1^H-NMR signals at 7.06 (1H, s, S-H), 9.2 (N=CH), 9.9–10.17 (C=NH), ^13^C-NMR signals at 193 (N=C–S), and the appearance of mass signals at 624 and 530.

### 3.2. Biological Activity

#### 3.2.1. In Vitro Antibacterial Activity of Synthesized Compounds

In vitro antibacterial assay consists of numerous biological assays such as agar dilution, well-diffusion, disk-diffusion, and broth dilution methods [48]. The antibacterial screening of all synthesized **compounds 5a-e** and **6a**,**b** against *P. gingivalis* and *E.* coli resulted in a minimum inhibition concentration (MIC) between 31 μg/mL and 250 μg/mL (Table 1). However, not all the synthesized **compounds** yielded a result for minimum bactericidal concentration (MBC). The synthesized **compounds 5b**, **5d**, **5e**, and **6b** have all displayed an MBC against *P. gingivalis* with a range of 250 μg/mL to 125 μg/mL. While for *E. coli,* only **compounds 5c**, **5e**, and **6b** have yielded MBC results with a range of 125 μg/mL to 250 ug/mL (Table 2). Moreover, both **compounds 5a** and **6a** displayed no MBC activity against *P. gingivalis* and *E. coli*, respectively. On the contrary, **compounds 5e** and **6b** are the only two **compounds** to yield MBC against both *P. gingivalis* and *E. coli.*

The synthesis of oxadiazole **compounds** in recent years has spiked up tremendously, solely due to their biological activities. Their antibacterial activity against pathogenic microorganisms has exceeded some of the known antibiotics, making them an alternative to combat drug resistance organisms [31]. It was previously reported by [49], that evaluation of scaffold oxadiazoles has resulted in MIC against gram-positive and gram-negative organisms. In addition, multiple other studies have shown that oxadiazoles have suppressed bacterial growth at low concentrations [50,51,52,53], similar to the observation that was noted in this study. Although oxadiazoles have broad-spectrum antibacterial activities [54], and their activities have been evaluated in both gram-positive and gram-negatives some of which are associated with oral diseases [55,56], there is no data on its evaluation and inhibition activity against *P. gingivalis.* This study may be the first one to report the ability of oxadiazoles to inhibit *P gingivalis* growth, a putative organism that promotes PD.

In the continuous search for alternatives to antibiotics, triazole Schiff bases derivatives possess the biological properties that inhibit the growth of drug-resistant organisms [57]. In the present study, it is worth knowing that **compound 6b** has yielded MIC and MBC against both *P. gingivalis* and *E. coli* compared to other synthesized compounds (Table 1 and Table 2). But this is not a surprise considering its known activity against pathogenic organisms [58], with some studies reporting that it is twice as active as ciprofloxacin [59], whereas other studies reported that it has activity comparable to that of chloramphenicol [60]. Furthermore, it was reported by [61,62], that triazole Schiff bases have an inhibition activity against multi-drug resistance organisms. Despite triazole Schiff base’s diverse antibacterial activity, there’s less evaluation of its activities against *P. gingivalis.* Nevertheless, a study showed that synthesized triazoles have the activity to inhibit adherence of *P. gingivalis* [30]. Another study showed that triazole has an inhibitory influence against *HmuY* and *fimA* gene expression (hemin binding proteins) which are responsible for *P. gingivalis* growth [63]. In the present study, all synthesized compounds have yielded antibacterial effects against tested pathogens with some having higher potential than others, making them a promising therapeutic alternative. Although all synthesized compounds exhibited significant activity against *P. gingivalis* and *E. coli*; however, among all, **compound 6b** was found to be most active, as it exhibited the best MIC and MBC values against *P. gingivalis* and *E. coli.*

#### 3.2.2. Cytotoxicity Analysis of Synthesized Compounds

In drug development, cytotoxicity is an important aspect of biological evaluation. In the present study, the MTT assay of all synthesized compounds was evaluated against HEK 293 cells. A previous study suggested that oxadiazole compounds yielded no to less toxicity when tested against NIH/3T3 cells due to cell viability greater than 75% [64]. Testing synthesized compounds of oxadiazoles against A549, L929, and HpG2 cells [65], showed that most synthesized compounds yielded no cytotoxicity against tested cell lines with cell viability greater than 75%, with only a few compounds resulting in cell death. Satisfactory results and minimal cytotoxicity of oxadizoles evaluation were previously described [66,67], which is in agreement with this study. In the current study, it was observed that all synthesized oxadiazoles (**5a-e**) yielded more than 70% cell viability when tested against HEK 293 cells at 62.5 μg/mL (Table 3 and Figure 1). At 125 μg/mL, all synthesized oxadiazoles showed no sign of cytotoxicity, except for **5b**, having only 66% cell viability. At the maximum concentration tested (500 μg/mL), only **compound 5d-e** showed no signs of cytotoxicity compared to other synthesized **compounds** (**5a-c**). Hence the minimal cytotoxicity at high concentrations and high antibacterial at low concentrations of these synthesized compounds are their major advantages.

Triazole has exemplary biological activity and due to its minimal cytotoxicity, it is ideal for many biological studies. Triazole Schiff bases have shown to be safe and have relatively less cytotoxicity when tested against HEK-293 and WI-38, respectively [68,69,70]. Triazole Schiff bases analysis against kidney, red blood cells, and lung cells all yielded no cytotoxicity, hence supporting their minimal cytotoxicity, and it is even suggested to be safer than cisplatin [71,72,73]. Here in this study, it was observed that **6a-b** resulted in more than 70% cell viability when tested against HEK-293 cells at 62.5 μg/mL (Table 3 and Figure 1). However, at 250 μg/mL, only **6b** resulted in more than 70% cell viability. Among **6a** and **6b**, the **compound 6b** is found to be much safer with 68% cell viability (Table 3) at the maximum tested concentration (500 μg/mL). 

Based on the resultant MIC data of synthesized **compounds 5a-e** and **6a**,**b** given in Table 1, the structure of benzamidine and its analogues synthesized in the present study, were related to their inhibitory potential (MIC) against PD triggering bacteria. The study revealed that incorporation of heterocyclic ring (oxadiazole and triazole) increases their inhibitory potential by twofold against *P. gingivalis* in comparison to parent benzamidine **compound 1**. It is observed that incorporation of Cl at ortho, NO_2_ at ortho and meta, and NH_2_ group at para position of benzene ring that is directly attached to oxadiazole ring containing benzamidine analogues **5b**, **5d**, and **5e**, enhances their inhibitory potential against *P. gingivalis* in comparison to **compound 1**. However, incorporation of Cl at para position of benzene ring in **compound 5a** offers activity similar to **compound 1**. Whereas incorporation of NO_2_ group at para position on benzene ring attached to triazole containing benzamidine analogue **6b** further enhances their inhibitory potential against *P. gingivalis* in comparison to **compound 1**. The resultant MBC data of **compound 5b**, **5d**, **5e**, and **6b** given in Table 2, revealed their equipotent MBC value when compared with parent **compound** 1. As per the resultant cyto-toxicity study data given in Table 3 and Figure 2, the **compounds 5a**, **5c-e**, and **6b** can be considered as nontoxic and safer alternatives for the treatment of PD. However, **compound 5a** containing para substituted Cl on benzene and **compound 6a** containing phenoxy group at meta position of benzene offers lesser safety when compared with other synthesized compounds. Free hydroxy group is not essential for the activity, conversion into ether linkage further enhances the inhibitory activity. Based on the MIC, MBC, and cytotoxicity data it is recommended that these synthesized compounds should be further subjected to preclinical and clinical evaluation.

## 4. Conclusions

In conclusion, compounds were successfully synthesized by the condensation of hydrazides with different aromatic benzoic acids, and cyclo-condensation of a triazole with imine Schiff bases. The synthesized compounds were further confirmed based on sharp melting point, single spot TLC pattern, and spectral data. All synthesized compounds displayed minimum to high inhibition activity against tested pathogens. Addiontally, all the synthesized compounds showed less cytotoxicity when tested against HEK-293 cells. Despite the present study showing the ability of benzamidine derivatives to inhibit the growth of periodontal pathogens with an absence of cytotoxicity of some of these derivatives, additional in-vivo and clinical studies are required to establish their safety and efficacy.

## Figures and Tables

**Figure 1 antibiotics-12-00306-f001:**
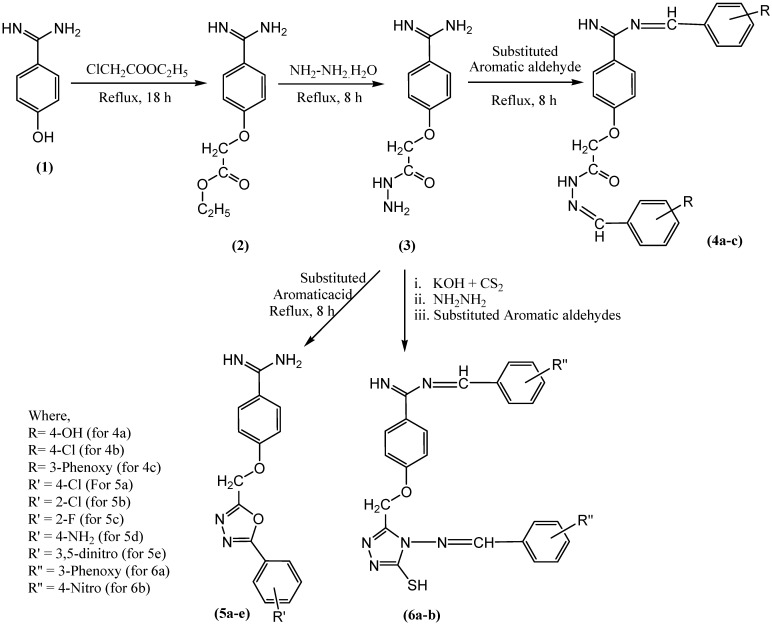
Scheme for synthesis of novel benzamidine analogues.

**Figure 2 antibiotics-12-00306-f002:**
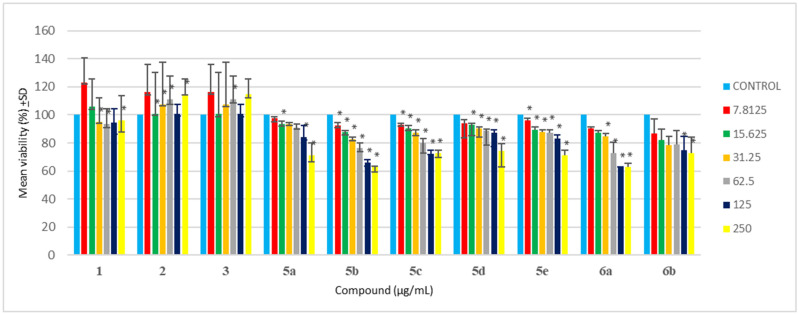
Cytotoxicity analysis of synthesized compounds against HEK-293 cells (Where, * *p* < 0.05).

**Table 1 antibiotics-12-00306-t001:** MIC values of synthesized **compounds**.

Compound (μg/mL)	Organisms	
	*P. gingivalis*	*E. coli*
**1**	62.5 ± 0.00 ^c^	31.25 ± 0.00 ^b^
**2**	62.5 ± 0.00 ^c^	55.5 ± 12.03 ^c^
**3**	62.5 ± 0.00 ^c^	31.5 ± 0.00 ^b^
**5a**	62.5 ± 0.44 ^c^	250 ± 1.76 ^d^
**5b**	31.25 ± 0.11 ^b^	250 ± 0.00 ^d^
**5c**	62.5 ± 0.00 ^c^	250 ± 1.76 ^d^
**5d**	31.25 ± 0.00 ^b^	250 ± 0.00 ^d^
**5e**	31.25 ± 0.88 ^b^	250 ± 1.76 ^d^
**6a**	125 ± 0.00 ^d^	250 ± 0.00 ^d^
**6b**	31.25 ± 0.00 ^b^	62.5 ± 0.00 ^c^
Ampicillin	15.63 + 0.00 ^a^	1.600 + 0.00 ^a^

MIC: Minimum inhibitory concentration. Data presented as mean ± standard error of each experiment performed in triplicate values. Means with different superscripts (a–d) were significantly different (p < 0.05).

**Table 2 antibiotics-12-00306-t002:** MBC values of synthesized **compounds**.

Compound (μg/mL)	Organisms	
	*P. gingivalis*	*E. coli*
**1**	125	-
**2**	125	-
**3**	125	-
**5a**	-	-
**5b**	125	-
**5c**	-	250
**5d**	125	-
**5e**	125	250
**6a**	-	-
**6b**	125	125
Ampicillin	62.5	7.8

MBC: Minimum bactericidal concentration.

**Table 3 antibiotics-12-00306-t003:** Cytotoxicity values of synthesized compounds.

Concentration (µg/mL)	Cell Viability (%)									
	1	2	3	5a	5b	5c	5d	5e	6a	6b
7.8125	122.87 ± 17.59	116.06 ± 19.78	116.06 ± 19.78	97.67 ± 0.90	92.67 ± 1.62 *	92.83 ± 1.35 *	96.00 ± 2.52	93.87 ± 1.78 *	90.53 ± 1.00	86.83 ± 10.09
15.625	105.89 ± 19.53	100.67 ± 29.35	100.67 ± 29.35 *	93.33 ± 2.08	87.93 ± 0.87 *	90.10 ± 2.19 *	89.13 ± 1.35 *	92.80 ± 2.12 *	87.47 ± 1.35 *	82.13 ± 7.81
31.25	94.83 ± 17.46 *	107.65 ± 29.99 *	107.65 ± 29.99	93.53 ± 0.84	82.77 ± 1.40 *	87.37 ± 1.85 *	88.00 ± 1.04 *	90.23 ± 1.19 *	84.83 ± 2.15 *	78.20 ± 6.30 *
62.5	93.56 ± 10.72 *	111.10 ± 16.38 *	111.10 ± 16.38 *	91.03 ± 2.66	76.10 ± 3.72 *	80.23 ± 2.83 *	87.00 ± 1.23 *	88.57 ± 2.11 *	72.80 ± 7.50 *	78.77 ± 10.12 *
125	94.41 ± 9.93	100.71 ± 6.74	100.71 ± 6.74	84 ± 8.22 *	66 ± 1.69 *	72 ± 2.15 *	83 ± 2.05 *	87 ± 2.51 *	62 ± 0.86 *	74 ± 9.77 *
250	96.03 ± 17.65 *	114.80 ± 10.81 *	114.80 ± 10.81	71 ± 8.42 *	62 ± 0.70 *	72 ± 2.48 *	71 ± 4.86 *	74 ± 3.71 *	62 ± 3.06 *	72 ± 11.41 *
Control	100 ± 0.00	100 ± 0.00	100 ± 0.00	100 ± 0.00	100 ± 0.00	100 ± 0.00	100 ± 0.00	100 ± 0.00	100 ± 0.00	100 ± 0.00

Data presented as mean ± standard error with each experiment were performed in triplicate. Mean values having superscript ‘*’ statistically indicates by * *p* < 0.05.

## Data Availability

The data presented in this study are available on request from the corresponding author.

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
