# Peer review of "Synthesis, Characterization, Cytotoxicity Analysis and Evaluation of Novel Heterocyclic Derivatives of Benzamidine against Periodontal Disease Triggering Bacteria"

_antibiotics, 2023, doi:10.3390/antibiotics12020306_

Round 1
Reviewer 1 Report
The current manuscript presents an interesting study on the synthesis and characterization of novel molecules for periodontitis treatment. Is in overall well done, described and discussed. Nevertheless, some alterations should be done before acceptance for publication:
- There are some numbers/letters/brackets highlighted in bold in the abstract that should not be highlighted, since they do not add any meaning or facilitate a readers’ comprehension;
- Figures should not be called “Scheme”, but “Figure” instead; hence, “Scheme 1” should be “Figure 1”, and “Figure 1” should be “Figure 2”;
- The “Methods” section should come before the “Results and discussion” section;
- Some sentences of the “Results and discussion” section should move on to the “Methods” section, such as “In the present study, the micro-broth dilution method was used to determine the inhibition susceptibility of synthesized compounds against P. gingivalis (ATCC 33277) and E. 137 coli (ATCC 25922).”;
- Abbreviations should be defined in each Table’s footnote, such as “MIC” and “MBC”;
- The figure of the cytotoxicity analysis of synthesized compounds against HEK-293 cells does not include any statistical significance; were statistical tests not applied? If not, they should be;
- In the conclusion section, where you say “additional studies are required to confirm the mechanisms that resulted in these high potential biological activities”, which tests are you referring to? Could you specify?
Author Response
POINT BY POINT RESPONSE TO REVIEWERS
We would like to express our heartfelt gratitude to the reviewers for taking the time to read the manuscript and provide us with constructive feedback. All of your suggestions have been taken into consideration in order to improve the quality of our manuscript. All changes have been highlighted with sky-blue, green and yellow colours in the revised version of the manuscript. The following are detailed responses to each of the reviewer comments.
REVIEWER 1
The current manuscript presents an interesting study on the synthesis and characterization of novel molecules for periodontitis treatment. Is in overall well done, described and discussed. Nevertheless, some alterations should be done before acceptance for publication:
Reply: Authors are thankful for the reviewer comments for the modifications of this manuscript. The suggestions and comments given by the reviewer will definitely upgrade the standard of manuscript.
Modifications in the manuscript for each comment or suggestion have been highlighted with sky-blue colour in the manuscript.
No |
Comments and Response |
1 |
There are some numbers/letters/brackets highlighted in bold in the abstract that should not be highlighted, since they do not add any meaning or facilitate a readers’ comprehension.
Authors appreciate the reviewer’s comment. As per the comment now all highlights have been removed in the abstract part.
|
2 |
Figures should not be called “Scheme”, but “Figure” instead; hence, “Scheme 1” should be “Figure 1”, and “Figure 1” should be “Figure 2”;
As per the reviewer suggestion scheme 1 and figure 1 are modified as Figure 1 and figure 2.
|
3 |
The “Methods” section should come before the “Results and discussion” section.
Authors understand the reviewer concern. Actually, we followed the “Antibiotics-MDPI template”, in that method section is after results and discussion. But as suggested now the “Methods” section has been placed before the “Results and discussion” section.
|
4 |
Some sentences of the “Results and discussion” section should move on to the “Methods” section, such as “In the present study, the micro-broth dilution method was used to determine the inhibition susceptibility of synthesized compounds against P. gingivalis (ATCC 33277) and E. 137 coli (ATCC 25922).
As per the reviewer suggestion the “Results and discussion” and “Methods” section have been modified now.
|
5 |
Abbreviations should be defined in each Table’s footnote, such as “MIC” and “MBC.
Authors understand the reviewer’s concern. Authors would like to highlight that abbreviations have been defined before in the text given in lines 137-139 of section 2.2.1. But as suggested the same has been defined in the footnote also of each table 1 and table 2.
|
6 |
The figure of the cytotoxicity analysis of synthesized compounds against HEK-293 cells does not include any statistical significance; were statistical tests not applied? If not, they should be
Authors appreciate the reviewer’s comment. As per the comment the statistical analysis has been incorporated in the figure of cytotoxicity analysis. |
7 |
In the conclusion section, where you say, “additional studies are required to confirm the mechanisms that resulted in these high potential biological activities”, which tests are you referring to? Could you specify?
Authors understand and appreciate the reviewer concern. As required the suggested text in the conclusion section has been modified.
|
Reviewer 2 Report
I found this article interesting for the readers and followed the journal Antibiotics’ scope. I don’t have any major comments as this article has enough data, however, the author could have discussed SAR of benzamidine analogs to make this article more interesting to the reader of Antibiotics.
I would recommend the article be published in Antibiotics after minor corrections.
The author needs to address the following comments/corrections.
1. The author could change the title of the manuscript as it is misleading.
2. Abstract is too long and needs to be shortened.
3. Introduction needs to be shortened.
4. NMRs (1H and 13 C) spectra should be in SI.
5. In Scheme 1, reagents, and yield (reaction condition) should be included.
6. The author should write experimental details (e.g. crystallization condition, workup condition etc.).
7. The Author should have made a few more analogs with versatile functional groups.
8. Any finding about active moiety of the molecule responsible for the biological activity.
9. Any rationale for choosing the analogs for testing.
10. The author could include the following reference.
(a) Journal of Medicinal Chemistry 2008, 51(23), 7344-7347.
(b) Journal of Inorganic Biochemistry, Volume 210, September 2020, 111164.
Author Response
POINT BY POINT RESPONSE TO REVIEWERS
We would like to express our heartfelt gratitude to the reviewers for taking the time to read the manuscript and provide us with constructive feedback. All of your suggestions have been taken into consideration in order to improve the quality of our manuscript. All changes have been highlighted with sky-blue, green and yellow colours in the revised version of the manuscript. The following are detailed responses to each of the reviewer comments.
REVIEWER 2
I found this article interesting for the readers and followed the journal Antibiotics’ scope. I don’t have any major comments as this article has enough data, however, the author could have discussed SAR of benzamidine analogs to make this article more interesting to the reader of Antibiotics.
Reply: Authors are thankful for the reviewer comments and suggestion for the modifications of this manuscript. As required, we have incorporated the SAR of in the discussion section.
I would recommend the article be published in Antibiotics after minor corrections. The author needs to address the following comments/corrections.
Reply: Authors are thankful for the reviewer comments for the modifications of this manuscript. The suggestions and comments given by the reviewer will definitely upgrade the standard of manuscript. The response to reviewer comments and suggestions is given in the below table.
Modifications in the manuscript for each comment or suggestion have been highlighted with green colour in the manuscript.
No |
Comments and Response |
1 |
The author could change the title of the manuscript as it is misleading.
Authors appreciate the reviewer’s suggestion. As per the suggestion now the title has been modified.
|
2 |
Abstract is too long and needs to be shortened.
Authors understand the reviewer’s concern. As per the comment now the abstract has been shortened.
|
3 |
Introduction needs to be shortened.
Authors appreciate the reviewer’s comment. As per the comment now the introduction has been shortened.
|
4 |
NMRs (1H and 13 C) spectra should be in SI.
Authors understand the reviewer’s concern. As per the comment now the NMRs (1H and 13 C) data has been mentioned as per the format. |
5 |
In Scheme 1, reagents, and yield (reaction condition) should be included.
Authors understand the reviewer’s concern. As per the comment now the scheme has been modified as per the suggested format. |
6 |
The author should write experimental details (e.g. crystallization condition, workup condition etc.)
Authors understand the reviewer’s concern. As per the comment now further experimental details has been given. |
7 |
The Author should have made a few more analogs with versatile functional groups.
Authors appreciate the reviewer’s comment. The authors would like to highlight that this study is the extension of the previous study (below mentioned) published by them in Antibiotics-MDPI in the year 2002. · Sa’ad MA, Kavitha R, Fuloria S, Fuloria NK, Ravichandran M, Lalitha P. Synthesis, Characterization and Biological Evaluation of Novel Benzamidine Derivatives: Newer Antibiotics for Periodontitis Treatment. Antibiotics. 2022; 11(2):207.https://doi.org/10.3390/antibiotics11020207
In the above-mentioned study authors have synthesized some more analogues with some other functional groups. A description for the same has been given in the figure 1 (Scheme) and in the introduction part.
|
8 |
Any finding about active moiety of the molecule responsible for the biological activity.
Authors appreciate the reviewer’s comment. As per the comment now a short discussion over finding about active moiety of the molecule responsible for the biological activity has been incorporated in the results and discussion section 2.2.1 and section 2.2.2 (structural activity relationship)
|
9 |
Any rationale for choosing the analogs for testing.
Authors understand the reviewer concern. As required a description over rationale for choosing the analogues has been incorporated in the introduction part.
|
10. |
The author could include the following reference. (a) Journal of Medicinal Chemistry 2008, 51(23), 7344-7347. (b) Journal of Inorganic Biochemistry, Volume 210, September 2020, 111164.
Authors appreciate the reviewer’s comment. As per the comment the suggested references have been incorporated in the manuscript.
|
Reviewer 3 Report
The manuscript describes the synthesis and biological activity of novel derivatives of Benzamidine. It addresses one of the key issues ailing most people around the world. While the compounds were not very potent, they are not cytotoxic; thus, they provide a base for the scientific community for further exploration. But the authors gave sufficient background information throughout the manuscript, and I commend them for that.
Here are my observations/comments/suggestions:
1. The nature of the substituents on the benzene ring and its effect on activity/toxicity was not explained.
2. I would recommend the authors to optimize further and add a few more compounds to the manuscript, because there is scope for improvement.
3. The NMR data were not given in a proper format and 13C data look incomplete. Why was the supplementary information file not provided for the manuscript?
4. Both the calculated and the found values should be given for mass spectroscopy data including the decimals.
5. R' was not defined for 5e in Scheme 1, instead it was given twice for 5b.
6. Reaction conditions can be given on the scheme or as a text under the scheme to make it easier for readers.
Author Response
POINT BY POINT RESPONSE TO REVIEWERS
We would like to express our heartfelt gratitude to the reviewers for taking the time to read the manuscript and provide us with constructive feedback. All of your suggestions have been taken into consideration in order to improve the quality of our manuscript. All changes have been highlighted with sky-blue, green and yellow colours in the revised version of the manuscript. The following are detailed responses to each of the reviewer comments.
REVIEWER 3
The manuscript describes the synthesis and biological activity of novel derivatives of Benzamidine. It addresses one of the key issues ailing most people around the world. While the compounds were not very potent, they are not cytotoxic; thus, they provide a base for the scientific community for further exploration. But the authors gave sufficient background information throughout the manuscript, and I commend them for that.
Here are my observations/comments/suggestions:
Reply: Authors are thankful for the reviewer comments for the modifications of this manuscript. The suggestions and comments given by the reviewer will definitely upgrade the standard of manuscript. The response to reviewer comments and suggestions is given in the below table.
No |
Comments and Response |
1 |
The nature of the substituents on the benzene ring and its effect on activity/toxicity was not explained.
Authors appreciate the reviewer’s comment. As per the comment now the the nature of the substituents on the benzene ring and its effect on activity/toxicity has been explained in the results and discussion section.
|
2 |
I would recommend the authors to optimize further and add a few more compounds to the manuscript, because there is scope for improvement.
Authors understand the reviewer’s concern. The authors would like to highlight that this study is the extension of the previous study (below mentioned) published by them in Antibiotics-MDPI in the year 2002. · Sa’ad MA, Kavitha R, Fuloria S, Fuloria NK, Ravichandran M, Lalitha P. Synthesis, Characterization and Biological Evaluation of Novel Benzamidine Derivatives: Newer Antibiotics for Periodontitis Treatment. Antibiotics. 2022; 11(2):207.https://doi.org/10.3390/antibiotics11020207
In the above-mentioned study authors have synthesized some more analogues with some other functional groups.
Now a description for the same has been given in figure 1 (Scheme) and in the introduction part.
|
3 |
The NMR data were not given in a proper format and 13C data look incomplete. Why was the supplementary information file not provided for the manuscript?
Authors appreciate the reviewer’s comment. As required the NMR data now has been incorporated as per the MDPI format. As needed we have attached the supplementary file for the manuscript.
|
4 |
Both the calculated and the found values should be given for mass spectroscopy data including the decimals.
Authors understand the reviewer’s concern. As required now both the calculated and the found values for mass spectroscopy data is now mentioned in the manuscript. |
5 |
R' was not defined for 5e in Scheme 1, instead it was given twice for 5b.
Authors are thankful for the reviewer’s comment. It was a typo error, as required now the error has been corrected in the manuscript. |
6 |
Reaction conditions can be given on the scheme or as a text under the scheme to make it easier for readers.
Authors understand the reviewer’s concern. As per the comment now the scheme has been modified as per the suggested format. |